# Efficacy and Equivalency of Phytase for Available Phosphorus in Broilers Fed an Available Phosphorus-Deficient Diet

**DOI:** 10.3390/ani14010041

**Published:** 2023-12-21

**Authors:** Myunghwan Yu, Elijah Ogola Oketch, Jun-Seon Hong, Nuwan Chamara Chathuranga, Eunsoo Seo, Haeeun Park, Bindhu Lakshmibai Vasanthakumari, Hans Lee, Jung-Min Heo

**Affiliations:** 1Department of Animal Science and Biotechnology, Chungnam National University, Daejeon 34134, Republic of Korea; tomymh@cnu.ac.kr (M.Y.); elijahogolah@gmail.com (E.O.O.); gospel0342@naver.com (J.-S.H.); nuonechathuranga@gmail.com (N.C.C.); sksjang1@naver.com (E.S.); qkrgodms2@naver.com (H.P.); 2Kemin Industries Inc., Des Moines, IA 50317, USA; b.l.vasanthakumari@kemin.com; 3Kemin Animal Nutrition and Health, Asia Pacific, Seongnam 13631, Republic of Korea; hans.lee@kemin.com

**Keywords:** available phosphorus, broiler, equivalency value, low-phosphorus diet, phytase, tibia

## Abstract

**Simple Summary:**

Continuous concerns persist regarding skeletal disorders and their associated welfare issues in modern fast-growing broiler chickens. As an essential and the third most expensive nutrient in the poultry diet, phosphorus plays a crucial role in bone growth, and the strength and rigidity of the skeleton. About 60% of dietary phosphorus is utilized by poultry, with the remaining portion being excreted and potentially contributing to pollution when released into the environment. Hence, it was postulated that a multi-phased approach involving the reduction in phosphorus content in the feed, coupled with increased phosphorus utilization through phytase supplementation, could alleviate the economic burden associated with both phosphorus excretion and feed costs. Therefore, this study was conducted to assess the efficacy of phytase on the performance, carcass traits, nutrient digestibility, tibia characteristics, and inositol phosphorus degradation in broiler chickens fed an available phosphorus-deficient diet. The results indicate that a reduction in the available phosphorus to 0.20% with phytase inclusion initiated phytate degradation and, as a result, improved the productive performance, nutrient digestibility, and tibia traits of the broilers. These findings support the application of low-phosphorus diets with phytase supplementation in the poultry industry.

**Abstract:**

This study was conducted to assess the effectiveness of phytase on the performance, carcass traits, nutrient digestibility, tibia characteristics, and inositol phosphorus (IP) degradation in broiler chickens. Additionally, the available phosphorus (AP) equivalency of phytase in AP-deficient diets was estimated for 35 days after hatching. A total of 336 one-day-old Ross 308 broiler chicks were allocated to one of seven dietary treatments with six replications with eight birds per cage. The dietary treatments were as follows: (1) positive control containing 0.45% AP of the starter and 0.42% AP of the grower diet (PC), (2) 0.10% AP deficiency from the PC (NC-1), (3) 0.15% AP deficiency from the PC (NC-2), (4) 0.20% AP deficiency from the PC (NC-3), (5) NC-3 +phytase (500 FTU/kg; NC-3-500), (6) NC-3 + phytase (1000 FTU/kg; NC-3-1000), and (7) NC-3 + phytase (1500 FTU/kg; NC-3-1500). On d 35, the NC-3 diet exhibited lower tibia weight compared to the other treatments (*p* < 0.001). The NC-3-1500 group had higher calcium and phosphorus contents in the tibia than the other treatments on d 35 (*p* < 0.01). Phytase supplementation led to a reduced IP_6_ concentration and increased IP_3_ concentrations in different sections of the gastrointestinal tract on d 21 and 35 compared to the control diet (*p* < 0.05). In conclusion, based on the tibia phosphorus content, this study determined that 500 FTU/kg phytase was equivalent to 0.377% and 0.383% AP in the diet on d 21, and 0.317% and 0.307% AP in the diet on d 35, respectively. Likewise, 1000 FTU/kg was determined to be equivalent to 0.476% and 0.448% AP on d 21, and 0.437% and 0.403% AP on d 35, respectively. Furthermore, 1500 FTU/kg was determined to be equivalent to 0.574% and 0.504% AP on d 21, and 0.557% and 0.500 AP on d 35, respectively.

## 1. Introduction

Alongside energy and protein supply, the provision of phosphorus (P) is of critical importance to poultry diets, since it is the third most expensive nutrient. Phosphorus is involved in bone mineralization, energy metabolism, and nucleic acid synthesis and, subsequently, has an impact on the birds’ overall growth performance [1,2]. According to [3], the plant seeds used in corn and soybean meal diets for poultry contain approximately 50-80% of their total phosphorus in the form of phytate [Myo-inositol (1, 2, 3, 4, 5, 6) hexakisphosphate; IP_6_]; a compound that is largely ingestible by non-ruminant animals. Ref. [4] noted that phytate, a compound found in plant-based feedstuffs, can have anti-nutritional effects by increasing endogenous nutrient losses and forming insoluble complexes through nutrient binding and mineral chelation, which can reduce the bioavailability of important nutrients and minerals, such as calcium, phosphorus, and zinc.

Phytase, known as Myo-inositol hexakisphosphate phosphohydrolase, is an enzyme capable of decomposing nutrient-binding phytate into inositol penta-, tetra-, tri-, di-, and monophosphate (IP_5_ through IP_1_). This enzymatic process releases one phosphate molecule at a time in a sequential manner [5]. Because IP_6-3_ have the anti-nutritional effect of reducing nutrient absorption in monogastric animals, it is important to reduce the presence of IP_6-3_ by using phytase [6]. The overall target is the complete breakdown of IP_6_, IP_4_, and IP_3_, and to release inositol. In this regard, supplementation with phytase generally increases P utilization in poultry, leading to improved poultry productivity through several specific metabolic pathways. Furthermore, it reduces the reliance on inorganic phosphates in the diet and minimizes P excretion into the environment [2,7]. Therefore, accurately estimating the efficacy of phytase is crucial for optimizing its application to modern poultry diets.

The precise estimation of the available phosphorus (AP) concentrations with the dietary inclusion of phytase is challenging due to various factors, such as phytase sources, varying supplemental levels, and the interactions between nutrient components [8]. Numerous studies have assessed the AP equivalency of different phytase sources in broiler diets [9,10,11]. Nevertheless, there is a lack of experiments investigating the AP equivalency of phytase in broiler diets using monocalcium phosphate (MCP) as the inorganic P source. Hence, the purpose of this study was to compare the effects of different AP concentrations and phytase levels in the diets on growth performance, carcass traits, nutrient digestibility, tibia traits, and phytate degradation. This study also investigated the AP equivalency of phytase by evaluating its relationship to the tibia phosphorus concentration of broilers fed a typical corn–soybean meal.

## 2. Materials and Methods

### 2.1. Birds and Housing

A total of 336 one-day-old male Ross 308 broiler chickens (41.16 ± 0.23 g) were obtained from a local hatchery (Dongsan hatchery, Cheonan, Republic of Korea) and used in the current 35 d study. The birds were individually weighed and allocated to cages in a completely randomized design. Each battery cage (76 × 61 × 46 cm^3^) housed eight birds. The cages were equipped with two nipple drinkers and a metal trough to provide water and feed efficiently. The experimental diets were offered *ad libitum*, and the birds had continuous access to clean drinking water through the nipple drinkers. The birds were vaccinated against infectious bronchitis and Newcastle disease at the hatchery. The room temperature was maintained at 30 ± 1 °C from day 1 to 3, and then gradually decreased to 25 ± 1 °C until d 14 of age. Thereafter, a 24 ± 1 °C temperature was maintained throughout the experiment according to the Ross 308 broiler management guidelines [12]. The birds were allowed continuous lighting during the entire experimental period. 

### 2.2. Experimental Design and Diets

The birds were assigned to one of seven dietary treatments in a completely randomized design, with six replicate pens per treatment. Phygest HT, an *Escherichia coli*-derived 6-phytase (Kemin, Industries Asia Pte, Senoko Drive, Singapore) was used as the phytase source in this study. The experimental diets were formulated to evaluate the effects of dietary phosphorus deficiency and phytase supplementation. The experimental diets consisted of various formulations to evaluate the effects of AP deficiency and phytase supplementation. The dietary treatments included a nutritionally adequate positive control diet (PC), with appropriate levels of starter (day 1–21) and grower (day 22–35) diets containing 0.45% and 0.42% AP, respectively. To induce phosphorus deficiencies, three negative control diets (NC-1, NC-2, NC-3) were formulated with decreasing levels of AP compared to the PC (0.10% deficiency, 0.15% deficiency, and 0.20% deficiency, respectively). Furthermore, the NC-3 was supplemented with phytase at three different inclusion levels: NC-3-500 (500 FTU/kg), NC-3-1000 (1000 FTU/kg), and NC-3-1500 (1500 FTU/kg). The NC diets were carefully formulated to contain 0.35%, 0.30%, and 0.25% AP in the starter phase, and 0.32%, 0.27%, and 0.22% AP in the grower phase, while ensuring that the basal PC diet, which included corn and soybean meal, met or exceeded the nutritional specifications set by [13] and the Ross 308 nutrient recommendations [14]. The composition and calculated analysis of the starter and grower diets are presented in Table 1. The diets were provided in a mash form on an *ad-libitum* basis. Furthermore, Cr_2_O_3_ (chromium oxide powder, >99.9% purity, Sigma-Aldrich, St. Louis, MO, USA) was added as an indigestible marker for digestibility analysis at a proportion of 0.3% to all the experimental diets.

### 2.3. Growth Performance Measurements

The initial body weights of the birds were recorded upon arrival, and subsequent body weights (BW) were measured weekly (d 7, 14, 21, 28, and 35) throughout this study. The average daily gain (ADG) was calculated using the BW data. Feed intake was measured weekly based on the feed consumption in the individual cages. The mortality-corrected average daily feed intake (ADFI) and feed conversion ratio (FCR) were calculated for each cage for each week of the experiment. 

### 2.4. Post-Mortem Procedure and Sample Collection

On d 21 and 35, six birds per treatment (one bird per cage) that had body weights closest to the mean were selected and euthanized using carbon dioxide asphyxiation for sample collection. The dressing percentage of meat with giblets (heart, gizzard, and liver) was determined by dividing its weight by the live weight of the birds. The drumsticks (skinless), breast meat, and the right tibias were removed from the carcasses and weighed. The percentages of the breast meat, drumstick, and right tibia were calculated relative to the weight of the entire carcass. Subsequently, the right tibia was de-fleshed and dried for further analysis.

Abdominal incisions were made on each euthanized bird, and then the duodenum, jejunum, and ileum were separated from the gastrointestinal tract. The ileum was defined as the segment of the small intestine that extends from Meckel’s diverticulum to the ileocecal junction. The digesta of the ileum, as well as the duodenum/jejunum from birds subjected to the same treatment, was gently flushed with distilled water into labeled plastic containers and stored at −80 °C until further analysis for inositol phosphate degradation. Furthermore, the ileal digesta of the birds was also obtained for the evaluation of nutrient digestibility. 

### 2.5. Nutrient Digestibility

The apparent ileal digestibility (AID) of the dry matter (DM), crude protein (CP), gross energy, crude ash, calcium, and phosphorus contents were determined to estimate the rate of nutrient digestibility at the terminal ileum. Previously collected ileal digesta samples were thawed, dried at 55 °C for 24 h, ground, and strained through a 0.75 mm sieve (ZM 200 Ultra-Centrifugal Mill; Retsch GmbH & Co., KG, Haan, Germany). Standard procedures [15] were employed for the analysis of the nutrient fractions, while the concentration of chromic oxide was determined using the method of [16]. The AID was then calculated as follows: $$\mathrm{AID}\;(\%)=100-(100\times \frac{M_{diet}\times N_{digest}}{M_{digest}\times N_{diet}})$$

M_diet_ is the marker concentration in the diet, N_digest_ is the nutrient concentration in the ileal digesta, M_digest_ is the marker concentration in the ileal digesta, and N_diet_ is the nutrient concentration in the diet.

### 2.6. Tibia Characteristics

The right tibia bone from the eviscerated birds was dehydrated in ethanol for 72 h and defatted in a diethyl ether: methanol mixture (9:1) for 72 h. Subsequently, the samples were dried overnight at 105 °C and then ashed at 600 °C overnight in porcelain crucibles (AOAC method 942.05). The resulting weight of the tibia ash was quantified as grams of ash per 100 g dry fat-free tibia. The analysis of calcium (AOAC method 927.02) and phosphorus (AOAC method 965.17) in the ashed tibia bones was carried out following standard procedures [15].

### 2.7. Inositol Phosphate Degradation Assays

The digesta collected from the duodenum/jejunum and ileum, amounting to 0.5 mL, was filtered in buffer, and 20 μL or 40 μL with or without added internal IP_2-6_ standards at 1.5 mM were injected into the column. Myo-inositol phosphate esters IP_1-6_ standards (Sichem and Sigma-Aldrich, St. Louis, MO, USA), prepared in buffer G were also analyzed under the same conditions, with an injection volume of 40 μL. The linear range for IP_6_ was 10 to 250 μg in an injection volume of 40 μL. Filtrates were analyzed using high-performance ion chromatography and UV detection at 290 nm after post-column derivatization. IPs with different degrees of phosphorylation (IP_3-6_) and their positional isomers were separated, without enantiomer differentiation, onto a Carbo Pac PA 200 column and corresponding guard column. An Fe(NO_3_)_3_ solution with 0.1% Fe(NO_3_)_3_ · 9H_2_O in 2% HClO_4_ was used as the reagent for derivatization according to [17]. The elution order of the IP isomers was established using commercial standards. Peaks were detected, and their retention times corresponded to the retention times of the IP standards. IP_6_ was used for quantification, and the correction factors for differences in the detector responses for IP_3-5_ were used according to [18]. The limit of detection was defined for a signal:noise ratio of 3:1, and was 0.1 μmol/g of DM for IP_3-4_ isomers and 0.05 μmol/g of DM for IP_6_. The limit of quantification was defined for a signal:noise ratio of 6:1.

### 2.8. Statistical Analysis

The data obtained from the experiment were analyzed using the general linear model (GLM) procedure for one-way ANOVA in Statgraphics Centurion Version 18.1.12 software. A completely randomized design was employed for the analysis. In terms of the growth performance measurements, the experimental unit was defined as the cage. For carcass trait weights, digestibility, tibia composition, and inositol degradation, selected individual birds were considered as the experimental unit. Statistical significance was determined at a significance level of *p* < 0.05. Whenever treatment effects were found to be significant (*p* < 0.05), the means were further analyzed and compared using Tukey’s multiple range test procedures implemented in Statgraphics Centurion. 

Linear and quadratic regression analyses were performed to test the effect of the increasing levels of phytase, and were also conducted to calculate the AP equivalency values of phytase, based on the concentrations and total amounts of phosphorus in the tibia as the response criteria, using the polynomial regressions described by [19]. The regression equations for the AP levels in the diet and supplemental phytase levels for particular response variables were equated and solved for x.
Y_s_ = a_s_ + b_s_x_s_ (linear regression equation for AP)(1)
Y_P_ = a_p_ + b_p_x_p_ (linear regression equation for phytase)(2)
Equation (1) = Equation (2): a_s_ + b_s_x_s_ = a_p_ + b_p_x_p_
where Y is the response criterion (concentrations of tibia phosphorus), x_s_ is the AP level in the diet, x_p_ is the supplemented phytase, a_s_ is the intercept of the linear regression for AP levels in the diets, a_p_ is the intercept of the linear regression for the supplemented phytase, b_s_ is the slope of the response criterion to the dietary AP levels, and b_p_ is the slope of the response criterion to the supplemented phytase. The linear response equations for the AP in the diet, and that for supplemented phytase, were set to be equal and were solved for the AP equivalency values for their respective variables.

The quadratic regression equation was also calculated in the same manner as the linear equation, as follows:Y_s_ = a_s_ + b_s_x_s_ + c_s_x_s_^2^ (quadratic regression equation for AP)(3)
Y_P_ = a_p_ + b_p_x_p_ + c_p_x_p_^2^ (quadratic regression equation for phytase)(4)
Equation (3) = Equation (4): a_s_ + b_s_x_s_ + c_s_x_s_^2^ = a_p_ + b_p_x_p_ + c_p_x_p_^2^

## 3. Results

### 3.1. Growth Performance

As presented in Table 2, the broilers fed the NC diets recorded lower (*p* < 0.001) BW compared to the PC and the phytase-supplemented NC-3-500, NC-3-1000, and NC-3-1500 diets for all the periods measured, except for day 1. Throughout the entire experimental period, the lowest BW values were recorded for the NC-3 diet (*p* < 0.001). No significant differences (*p* > 0.05) were observed in the BWs of the broilers, not only between the phytase-supplemented diets and the PC, but also among the different phytase levels for all periods. Likewise, lower ADG values were recorded for the NC diets compared to the PC and phytase-supplemented NC-3-500, NC-3-1000, and NC-3-1500 diets. Furthermore, no significant differences (*p* > 0.05) were noted in the ADG among the broilers fed the various phytase-supplemented diets compared to the PC during the starter (d 1–21) and over the entire period (d 1–35). Similarly, feeding the NC-3 diet resulted in lower (*p* < 0.001) average daily gains throughout the entire experimental period. The daily feed intake levels were unaffected by phosphorus reduction or phytase supplementation throughout the experiment. Considering the FCR, birds fed the NC-3 diet reported lower (*p* < 0.01) feed efficiency compared to the PC and phytase-supplemented diets during the entire period of the experiment (d 1 to 35). Moreover, there were no changes (*p* > 0.05) in the FCR compared to the PC and all the phytase diets during the starter (d 1 to 21) or the entire period of the experiment (d 1 to 35).

### 3.2. Carcass Traits

No significant differences (*p* > 0.05) were observed in the dressing percentages, and the relative breast meat and drumstick weights among the various dietary treatments on both d 21 and 35 are shown in Table 3. However, the broilers fed the NC-3 diet exhibited lower tibia weight (*p* < 0.001) compared to the other dietary treatments on d 35. On the other hand, there were no differences (*p* > 0.05) in tibia weight between the PC diet and all the phytase-supplemented diets on both d 21 and 35.

### 3.3. Nutrient Digestibility

The AID of energy was decreased (*p* < 0.001) in all the NC diets compared to the PC diet on d 21 and 35 (Table 4). The AID of energy in the phytase-supplemented diets was median to that of the PC and NC diets. The apparent CP digestibility was not affected (*p* > 0.05) for either the PC or NC-3 with phytase diets on d 21 and 35. However, the AID of energy on d 35 was improved (*p* < 0.001) in the birds fed the NC-3-1000 diet compared to all the treatments. Nonetheless, the broilers fed the NC-3 diet had a lower (*p* < 0.05) crude ash digestibility than those fed the PC or NC-3 with phytase diets on d 35. The NC-3-1500 diet increased (*p* < 0.05) the AID of phosphorus on d 21 and 35 compared to the NC diets.

### 3.4. Tibia Traits

Feeding NC-3 diets resulted in lower (*p* < 0.001) calcium and phosphorus contents in the tibia compared to the PC on d 21 (see Table 5). The tibia phosphorus content was higher (*p* < 0.001) in the broilers fed the NC-3-1500 compared to the other treatments on d 21 and 35. Furthermore, the birds fed the NC-3-1500 had higher (*p* < 0.01) contents of calcium on d 35 than those receiving the other treatments.

### 3.5. Inositol Phosphate Degradation

All the phytase-supplemented NC-3 diets recorded lower (*p* < 0.01) IP_6_ concentrations, not only in the duodenum/jejunum but also in the ileum, on d 21 and 35 compared to the PC diets (see Table 6). Regardless of the concentration, phytase supplementation in the NC-3 resulted in a higher (*p* < 0.001) IP_4_ concentration in the ileum on d 35 compared to the treatments without phytase. Birds fed the NC-3-1000 and NC-3-1500 diets had higher (*p* < 0.001) IP_3_ concentrations in the duodenum/jejunum, as well as in the ileum, on d 21 and 35 compared to the PC diets.

### 3.6. Regression Equations and Phosphorus Equivalency Values

The research findings regarding the relationship between increasing levels of dietary AP and supplemental phytase are summarized in Table 7. The calculated phosphorus equivalency values of the levels of phytase with NC-3 using linear and quadratic regressions are presented in Table 8. Based on the linear and quadratic regression models, 500 FTU/kg phytase was determined to be equivalent to 0.377% and 0.383% AP in the diet on d 21, and 0.317% and 0.307% AP in the diet on d 35, based on the concentrations of tibia phosphorus, respectively. Likewise, 1000 FTU/kg phytase was determined to be equivalent to 0.476% and 0.448% AP in the diet on d 21, and 0.437% and 0.403% AP in the diet on d 35, respectively. Finally, 1500 FTU/kg phytase was determined to be equivalent to 0.574% and 0.504% AP in the diet on d 21, and 0.557% and 0.500 AP in the diet on d 35, respectively.

## 4. Discussion

Several studies have evaluated the AP equivalency of phytase for a couple of inorganic sources (i.e., monosodium phosphate and dicalcium phosphate) to determine their effectiveness in diet formulation [9,11,19]. Despite having the highest P digestibility among inorganic P sources in broiler diets [1,20], there is a scarcity of available data regarding the AP equivalency of phytase with mono-calcium phosphate (MCP). The present research was designed to investigate the efficacy and AP equivalency of phytase on growth performance, carcass traits, nutrient digestibility, tibia bone mineralization, and inositol degradation in broilers.

In the current study, the growth performance of the broilers, except for the ADFI, was found to be reduced in all periods as the content of available phosphorus decreased. However, the body weight on d 35 increased by 17.86, 22.99%, and 20.98%, respectively, when 500, 1000, and 1500 FTU/kg of phytase were added to the NC-3. Likewise, the ADG for the entire experimental period improved by 18.28%, 23.54%, and 21.46%, respectively, for the diets containing 500, 1000, and 1500 FTU/kg of phytase relative to NC-3. Furthermore, the NC-3-500, NC-3-1000, and NC-3-1500 diets improved feed efficiency for the whole period by 17.86%, 18.45%, and 18.45%, respectively, compared to NC-3. The results of this current study are collectively in agreement with the findings of previous studies that have shown that phytase improves the growth performance of broilers fed available phosphorus-deficient diets [11,19,21]. Moreover, the observed decline in growth performance and feed efficiency with decreasing levels of dietary available phosphorus substantiates the inability of these birds to effectively utilize phytate phosphorus. This partial disagreement with previous studies could be due to differences in the composition of the feed ingredients and the type of inorganic phosphates.

With the exception of the tibia weights, the carcass traits were not influenced by a P reduction or phytase supplementation among the different treatments. An available phosphorus reduction without dietary phytase supplementation resulted in lowered tibia weights. The outcome of this study suggests the capacity of phytase supplementation to improve ileal mineral absorption and bone mineralization, which ultimately improves the tibia weight. Phytase supplementation could also correct the negative impact of low available phosphorus contents in the diets by improving the ileal Ca and P digestibility for improved tibia weights, as was observed. Although these results are consistent with [22], who found that supplementation with phytase did not affect the carcass traits, the research regarding the available phosphorus content and carcass traits remains insufficient.

In the current study, as the AP levels were decreased in the adequate PC diet, which was sufficient to meet nutrient requirements, the digestibility of the energy and phosphorus in the diets was negatively affected. According to [23], elevated dietary phytate concentrations result in diminished energy utilization and ileal digestibility of protein due to the inhibitory effects of phytate. The inhibitory action of elevated phytate levels results in nutrient-binding complexes that lower the hydrolyzing of endogenous enzymes and, thus, the availability of minerals and energy is hampered. In this present study, the digestibility of energy, phosphorus, crude protein, and ash on d 35 was increased in response to phytase supplementation in the reduced AP of diets. Increased nutrient digestibility is evidence of phytase activity resulting in the considerable breakdown of the phytate structure and the reduction in anti-nutritional factors. The breakdown of phytate into lower esters leads to a subsequent improvement not only of P, but could exert “extra-phosphoric effects” to improve the digestibility of energy, crude protein, and ash, as has been previously reported [4,24].

Due to its involvement in improving Ca and P digestibility and, thus, improved bone mineralization, it was also observed that dietary phytase supplementation improved the tibia characteristics, especially the total ash content, as has been previously reported [11,19,25]. While the current study is consistent with previous studies [11,19,25] regarding the tibia ash content, a similar trend was observed for the concentrations of calcium and phosphorus in the tibia. Likewise, ref. [9] demonstrated improvements in the tibia phosphorus content when increasing the graded levels of phytase in low-phosphorus diets. In line with the findings of [26], phytase supplementation contributes to increased phosphorus digestibility, consequently leading to higher concentrations of substrate available for tibia mineral deposition. This effect is supported by the observed increase in tibia ash and phosphorus contents in this study. However, ref. [21] noted that there was no difference in tibia phosphorus and calcium level related to the non-phytate phosphorus and phytase contents of the diets. The discrepancy between these results could be due to differences in the composition of the feed ingredients, inorganic phosphate types, and the age of the broilers.

It is expected that phytase supplementation facilitates a process in which phosphate molecules are released through a step-by-step dephosphorylation pathway, breaking down IP_6_ into lower esters, including IP4 and IP_3,_ which are also known to be antinutritive in nature [27]. The analysis of the inositol phosphate profiles in the duodenal/jejunal and ileal digesta from this experiment suggests that the highest proportion of IP_6_ concentrations are found in diets without phytase supplementation, particularly in the PC and NC diets (i.e., NC-1, NC-2, and NC-3). Subsequently, phytase supplementation in the NC-3-500, NC-3-1000, and NC-3-1500 diets results in the hydrolysis of IP_6_ into IP_5-3_ and lower esters. It has been reported [28] that phytase readily targets higher molecular weight esters to release IP_5_ more efficiently than the other esters. However, a focus should also be placed on improving the content of the lower esters, including IP_4_ and IP_3_. The superdosing of phytase as a concept is targeted at supplying higher phytase levels to result in as much IP_5_, IP_4,_ and IP_3_ as possible, as well as the eventual release of inositol. These findings are consistent with those of [6,29], who demonstrated a reduction in IP_6_ and an increase in IP_3_ in the ileum following dietary phytase supplementation. Going forward, the focus should be not only on the release of IP_4_ and IP_3_, but on their subsequent breakdown as well which results in the eventual release of inositol. Zinc, as a co-factor of pancreatic secretions, is known to be precipitated at higher pH levels in the ileum by IP_4_ and IP_3_, thus hindering protein degradation. This could be due to the pancreatic duct secreting zinc into the small intestine, which induces an increasing digesta pH and reduces the activity of phytase or, more plausibly, the impact of dietary phosphorus on the hydrolysis of phytate by exogenous phytase [30,31]. Unfortunately, we were not able to test this notion in the current study. 

## 5. Conclusions

Phytase supplementation resulted in a reduction in the IP_6_ concentration and a corresponding increase in IP_3_ concentrations in various sections of the GI tract at 21 and 35 days of age compared to the control diet. Based on the linear regression model, supplementation with 500, 1000, and 1500 phytase units per kg of diet was determined to be equivalent to 0.377, 0.476, and 0.574% AP on d 21, and 0.317, 0.437, and 0.557% on d 35, respectively. Additionally, employing the quadratic equation, supplementation with 500, 1000, and 1500 phytase units per kg of diet corresponded to 0.383, 0.448, and 0.504% AP on d 21, and 0.307, 0.403, and 0.500% on d 35, respectively. Our data consistently suggest that phytase can be used from 1 to 35 d of age in available phosphorus-deficient broiler diets to improve nutrient digestibility (Ca, P, energy, ash, and crude protein) and to stimulate phytate degradation into lower esters. Improved nutrient digestibility and phytate degradation are directly responsible for the improved growth performance and bone mineralization observed in the tibia in the current study.

## Figures and Tables

**Table 1 animals-14-00041-t001:** Composition (%, as-fed basis) of the experiment diets ^1^.

Ingredient (%)	Starter Phase (d 1–21)				Grower Phase (d 22–35)			
	PC	NC-1	NC-2	NC-3	PC	NC-1	NC-2	NC-3
Corn	50.94	51.52	51.79	52.06	55.02	55.61	55.88	56.19
Soybean meal, 45%	39.16	39.04	39.00	38.96	34.28	34.20	34.16	34.08
Limestone	1.42	1.60	1.69	1.78	1.46	1.63	1.72	1.81
Mono-calcium phosphate	1.48	1.04	0.84	0.60	1.40	0.96	0.72	0.52
Iodized salt	0.30	0.30	0.30	0.30	0.30	0.30	0.30	0.30
Beef tallow	5.36	5.16	5.04	4.96	6.32	6.08	6.00	5.88
DL-methionine, 98%	0.32	0.32	0.32	0.32	0.33	0.33	0.33	0.33
L-lysine-Sulfate, 65%	0.42	0.42	0.42	0.42	0.29	0.29	0.29	0.29
Vit-Min premix ^2^	0.30	0.30	0.30	0.30	0.30	0.30	0.30	0.30
Cr_2_O_3_	0.30	0.30	0.30	0.30	0.30	0.30	0.30	0.30
Calculated values								
Dry matter	88.17	88.10	88.06	88.03	88.23	88.16	88.12	88.09
ME, kcal/kg	3050	3050	3050	3050	3150	3149	3150	3149
Crude protein	23.0	23.0	23.0	23.0	20.0	20.0	20.0	20.0
Crude fiber	3.83	3.84	3.84	3.85	3.64	3.66	3.66	3.66
Calcium	0.90	0.90	0.90	0.90	0.88	0.88	0.88	0.88
Total phosphorus	0.71	0.60	0.56	0.50	0.66	0.56	0.51	0.46
Available phosphorus	0.45	0.35	0.30	0.25	0.42	0.32	0.27	0.22
Lysine	1.45	1.45	1.45	1.45	1.25	1.25	1.25	1.25
Methionine + Cysteine	0.99	0.99	0.99	0.99	0.95	0.95	0.95	0.95
Analyzed values								
Dry matter	88.84	88.88	88.79	88.51	88.60	88.76	87.84	88.06
Crude protein (Nitrogen × 6.25)	21.15	22.79	21.14	20.97	20.16	19.90	20.22	20.24
Gross energy, kcal/kg	4100	4052	4017	4063	4113	4152	4194	4182
Crude ash	6.58	6.27	6.33	6.59	6.15	6.10	6.15	6.61
Calcium	1.09	1.15	1.07	1.08	0.98	1.02	0.94	0.94
Total phosphorus	0.61	0.55	0.51	0.46	0.64	0.62	0.60	0.58

^1^ PC: Positive control diet contained the recommended calcium and available phosphorus; NC-1: negative control diet contained the recommended calcium and 0.10% available phosphorus deficiency; NC-2: negative control diet contained the recommended calcium and 0.15% available phosphorus deficiency; NC-3: negative control diet contained the recommended calcium and 0.20% available phosphorus deficiency. ^2^ Provided per kilogram of diet: vitamin A (trans-retinyl acetate), 14,000 IU; vitamin D3 (cholecalciferol), 3000 IU; vitamin E (DL-α Tocopherol acetate), 40 mg; vitamin K3, 2.4 mg; thiamin, 1.2 mg; riboflavin, 50 mg; pyridoxine, 3 mg; vitamin B12, 20 μg; niacin, 40 mg; pantothenic acid, 10 mg; folic acid, 0.5 mg; Fe (from iron sulfate), 17 mg; Cu (from copper sulfate), 13 mg; Zn (from zinc oxide), 92 mg; Mn (from manganese oxide), 100 mg; I (from potassium iodide), 1 mg; Co, 0.15 mg; Se (from sodium selenite), 0.25 mg.

**Table 2 animals-14-00041-t002:** Effect of phytase inclusion in diets on growth performance of broiler chickens ^1^.

Period	Dietary Treatment ^2^							SEM ^3^	*p*-Value	Polynomial Contrast ^4^	
	PC	NC-1	NC-2	NC-3	NC-3-500	NC-3-1000	NC-3-1500			Linear	Quadratic
BW, g											
Day 1	41.23	41.03	41.12	40.83	41.33	40.98	41.58	0.101	0.552	0.161	0.376
Day 7	121.70 ^cd^	113.17 ^ab^	113.88 ^abc^	108.07 ^a^	119.63 ^bcd^	122.67 ^d^	122.60 ^d^	1.302	0.004	0.003	0.002
Day 14	323.56 ^e^	270.75 ^abc^	263.01 ^ab^	245.72 ^a^	290.68 ^bcd^	301.69 ^cde^	304.35 ^de^	5.503	<0.001	0.001	0.001
Day 21	745.74 ^d^	620.50 ^bc^	565.17 ^ab^	537.61 ^a^	670.74 ^cd^	716.24 ^d^	710.18 ^d^	14.980	<0.001	<0.001	<0.001
Day 28	1336.75 ^c^	1148.91 ^ab^	1087.05 ^a^	1046.89 ^a^	1232.90 ^bc^	1326.34 ^c^	1302.83 ^c^	22.814	<0.001	<0.001	<0.001
Day 35	1989.76 ^c^	1825.97 ^b^	1771.90 ^ab^	1676.36 ^a^	1975.83 ^c^	2061.68 ^c^	2028.07 ^c^	27.831	<0.001	<0.001	<0.001
ADG, g/d											
Day 7	11.50 ^cd^	10.31 ^ab^	10.40 ^abc^	9.61 ^a^	11.19 ^bcd^	11.67 ^d^	11.57 ^d^	0.183	0.005	0.004	0.003
Day 14	28.84 ^d^	22.51 ^abc^	21.30 ^ab^	19.67 ^a^	24.44 ^bc^	25.57 ^cd^	25.96 ^cd^	0.647	<0.001	0.002	0.002
Day 21	60.31 ^d^	49.96 ^bc^	43.17 ^ab^	41.70 ^a^	54.29 ^cd^	59.22 ^d^	57.98 ^d^	1.427	<0.001	<0.001	<0.001
Day 28	84.43 ^bc^	75.49 ^ab^	74.55 ^ab^	72.75 ^a^	80.31 ^abc^	87.16 ^c^	84.66 ^bc^	1.524	0.047	0.010	0.013
Day 35	93.29 ^a^	96.72 ^ab^	97.84 ^abc^	89.93 ^a^	106.13 ^c^	105.05 ^cd^	103.61 ^bcd^	1.337	0.001	0.008	<0.001
Day 1–21	33.55 ^d^	27.59 ^bc^	24.96 ^ab^	23.66 ^a^	29.97 ^cd^	32.16 ^d^	31.84 ^d^	0.712	<0.001	<0.001	<0.001
Day 22–35	88.86 ^abc^	86.11 ^ab^	86.19 ^ab^	81.34 ^a^	93.22 ^bc^	96.10 ^c^	94.13 ^c^	1.208	0.004	0.004	0.001
Day 1–35	55.67 ^c^	51.00 ^b^	49.45 ^ab^	46.73 ^a^	55.27 ^b^	57.73 ^c^	56.76 ^c^	0.794	<0.001	<0.001	<0.001
ADFI, g/d											
Day 7	13.02	14.60	14.78	14.67	14.81	14.42	13.23	0.277	0.390	0.141	0.220
Day 14	33.92	36.02	36.09	37.86	33.84	34.52	33.48	1.173	0.961	0.475	0.725
Day 21	73.24	74.68	66.82	68.35	69.07	71.19	72.21	1.452	0.803	0.340	0.641
Day 28	109.44	113.83	110.70	116.09	109.63	123.32	119.79	2.370	0.653	0.401	0.692
Day 35	137.13	151.94	149.31	153.22	153.35	151.13	148.95	1.803	0.209	0.456	0.738
Day 1–21	40.06	41.77	39.23	40.29	39.24	40.04	39.64	0.782	0.987	0.913	0.985
Day 22–35	123.29	132.88	130.00	134.65	131.49	137.23	134.37	1.658	0.408	0.795	0.967
Day 1–35	73.35	78.21	75.54	78.04	76.14	78.92	77.53	0.985	0.795	0.913	0.989
FCR, g/g											
Day 7	1.14 ^a^	1.42 ^ab^	1.43 ^ab^	1.52 ^b^	1.36 ^abc^	1.25 ^ab^	1.14 ^a^	0.036	0.009	0.002	0.008
Day 14	1.19 ^a^	1.62 ^abc^	1.69 ^bc^	1.97 ^c^	1.41 ^ab^	1.36 ^ab^	1.26 ^ab^	0.065	0.010	0.013	0.022
Day 21	1.23 ^a^	1.53 ^ab^	1.64 ^b^	1.67 ^b^	1.27 ^a^	1.21 ^a^	1.25 ^a^	0.052	0.027	0.005	0.001
Day 28	1.30	1.52	1.54	1.63	1.37	1.42	1.43	0.043	0.485	0.265	0.265
Day 35	1.47	1.58	1.54	1.72	1.45	1.44	1.44	0.028	0.054	0.015	0.009
Day 1–21	1.21 ^a^	1.53 ^bc^	1.60 ^c^	1.72 ^c^	1.31 ^ab^	1.25 ^a^	1.24 ^a^	0.044	0.001	0.003	0.002
Day 22–35	1.39	1.55	1.53	1.68	1.41	1.43	1.43	0.030	0.108	0.046	0.030
Day 1–35	1.32 ^a^	1.54 ^bc^	1.54 ^bc^	1.68 ^c^	1.38 ^ab^	1.37 ^ab^	1.37 ^ab^	0.030	0.006	0.008	0.003

^1^ Values are mean of six replicates per treatment. ^2^ PC: Positive control diet contained the recommended Ca and non-phytate phosphorus; NC-1: negative control diet contained the recommended Ca and 0.10% available P deficiency; NC-2: negative control diet contained the recommended Ca and 0.15% available P deficiency; NC-3: negative control diet contained the recommended Ca and 0.20% available P deficiency; NC-3-500, NC-3-1000, and NC-3-1500 are NC-3 plus phytase 500, 1000, 1500 FTU/kg, respectively. ^3^ Pooled standard error of mean. ^4^ Orthogonal polynomial contrasts were conducted to assess the significance of the linear or quadratic effects of the supplementation with phytase in the broilers. ^a–e^ Values in a row with different superscripts differ significantly (*p* < 0.05).

**Table 3 animals-14-00041-t003:** Effect of phytase inclusion in diets on carcass traits of broiler chickens ^1^.

Period	Dietary Treatment ^2^							SEM ^3^	*p*-Value	Polynomial Contrast ^4^	
	PC	NC-1	NC-2	NC-3	NC-3-500	NC-3-1000	NC-3-1500			Linear	Quadratic
Dressing (%) ^5^											
Day 21	90.48	91.02	90.21	90.42	91.04	91.36	90.85	0.130	0.186	0.306	0.150
Day 35	92.64	91.31	92.22	92.15	92.96	92.22	92.27	0.169	0.255	0.824	0.578
Breast (%) ^6^											
Day 21	23.32	23.32	22.72	21.29	23.27	23.30	23.76	0.229	0.080	0.020	0.036
Day 35	27.44	26.42	25.86	24.29	25.68	27.02	26.42	0.284	0.067	0.030	0.041
Drumstick (%) ^7^											
Day 21	9.89	9.77	9.52	9.21	9.90	10.06	9.92	0.094	0.215	0.068	0.059
Day 35	10.54	10.00	10.15	9.95	10.13	10.14	10.75	0.087	0.118	0.006	0.012
Tibia weight (g)											
Day 21	5.83	5.50	4.83	4.67	5.33	5.33	5.50	0.124	0.158	0.078	0.158
Day 35	16.67 ^b^	15.17 ^ab^	15.00 ^ab^	12.50 ^a^	16.33 ^b^	16.67 ^b^	16.67 ^b^	0.307	<0.001	<0.001	<0.001

^1^ Values are mean of six replicates per treatment. ^2^ PC: Positive control diet contained the recommended Ca and non-phytate phosphorus; NC-1: negative control diet contained the recommended Ca and 0.10% available P deficiency; NC-2: negative control diet contained the recommended Ca and 0.15% available P deficiency; NC-3: negative control diet contained the recommended Ca and 0.20% available P deficiency; NC-3-500, NC-3-1000, and NC-3-1500 are NC-3 plus phytase 500, 1000, 1500 FTU/kg, respectively. ^3^ Pooled standard error of mean. ^4^ Orthogonal polynomial contrasts were conducted to assess the significance of the linear or quadratic effects of the supplementation with phytase in the broilers. ^5^ (Carcass weight/live body weight) × 100. ^6^ (Breast meat weight/carcass weight) × 100. ^7^ (Drumstick weight/carcass weight) × 100. ^a,b^ Values in a row with different superscripts differ significantly (*p* < 0.05).

**Table 4 animals-14-00041-t004:** Effect of phytase inclusion in diets on nutrient digestibility in broiler chickens ^1^.

Period	Dietary Treatment ^2^							SEM ^3^	*p*-Value	Polynomial Contrast ^4^	
	PC	NC-1	NC-2	NC-3	NC-3-500	NC-3-1000	NC-3-1500			Linear	Quadratic
Day 21 (%)											
Dry matter	64.20	64.84	64.73	64.83	64.36	64.71	64.51	0.397	0.999	0.711	0.906
Crude protein	80.72	81.05	79.33	79.50	79.95	80.87	79.95	0.214	0.180	0.452	0.417
Energy	79.32 ^e^	75.22 ^c^	72.70 ^b^	70.03 ^a^	76.03 ^cd^	77.06 ^d^	77.59 ^de^	0.502	<0.001	<0.001	<0.001
Ash	53.37	48.43	47.50	46.82	48.69	51.92	53.46	1.060	0.434	0.007	0.020
Calcium	53.67	52.37	52.32	52.53	53.98	56.20	55.10	1.058	0.955	0.219	0.331
Phosphorus	60.56 ^ab^	59.44 ^a^	58.93 ^a^	57.34 ^a^	59.64 ^a^	65.72 ^ab^	73.29 ^b^	1.342	0.011	0.001	0.004
Day 35 (%)											
Dry matter	76.03	76.22	75.96	75.36	76.68	76.08	75.67	0.382	0.988	0.950	0.775
Crude protein	89.09 ^cd^	87.62 ^bc^	86.95 ^ab^	85.32 ^a^	87.56 ^bc^	90.93 ^d^	88.93 ^c^	0.344	<0.001	0.001	<0.001
Energy	82.82 ^d^	79.07 ^b^	80.76 ^c^	77.43 ^a^	81.42 ^c^	84.51 ^e^	82.79 ^d^	0.368	<0.001	<0.001	<0.001
Ash	60.93 ^b^	56.68 ^ab^	56.63 ^ab^	52.89 ^a^	63.01 ^b^	63.25 ^b^	63.56 ^b^	1.047	0.021	0.032	0.032
Calcium	53.30	52.76	51.74	50.18	52.57	53.29	53.31	1.223	0.995	0.522	0.774
Phosphorus	63.16 ^abc^	60.15 ^a^	61.95 ^ab^	59.87 ^a^	64.02 ^abc^	65.16 ^bc^	66.67 ^c^	0.642	0.026	0.006	0.019

^1^ Values are mean of six replicates per treatment. ^2^ PC: Positive control diet contained the recommended Ca and non-phytate phosphorus; NC-1: negative control diet contained the recommended Ca and 0.10% available P deficiency; NC-2: negative control diet contained the recommended Ca and 0.15% available P deficiency; NC-3: negative control diet contained the recommended Ca and 0.20% available P deficiency; NC-3-500, NC-3-1000, and NC-3-1500 are NC-3 plus phytase 500, 1000, 1500 FTU/kg, respectively. ^3^ Pooled standard error of mean. ^4^ Orthogonal polynomial contrasts were conducted to assess the significance of the linear or quadratic effects of the supplementation with phytase in the broilers. ^a–e^ Values in a row with different superscripts differ significantly (*p* < 0.05).

**Table 5 animals-14-00041-t005:** Effect of phytase inclusion in diets on tibia traits of broiler chickens ^1^.

Period	Dietary Treatment ^2^							SEM ^3^	*p*-Value	Polynomial Contrast ^4^	
	PC	NC-1	NC-2	NC-3	NC-3-500	NC-3-1000	NC-3-1500			Linear	Quadratic
Ash (%)											
Day 21	50.24	44.42	43.67	41.37	46.69	47.59	51.48	1.053	0.120	0.007	0.028
Day 35	50.04 ^bc^	48.26 ^abc^	46.46 ^ab^	44.50 ^a^	48.38 ^bc^	48.73 ^bc^	52.03 ^c^	0.581	0.012	<0.001	0.003
Ca (%)											
Day 21	38.18 ^c^	32.65 ^ab^	30.28 ^a^	28.20 ^a^	37.18 ^bc^	37.88 ^bc^	41.83 ^c^	1.500	<0.001	<0.001	<0.001
Day 35	35.33 ^b^	33.08 ^ab^	31.97 ^ab^	30.28 ^a^	33.37 ^ab^	34.31 ^b^	41.62 ^c^	0.687	<0.001	<0.001	<0.001
P (%)											
Day 21	17.48 ^c^	14.40 ^ab^	12.90 ^a^	12.66 ^a^	16.12 ^bc^	16.54 ^bc^	20.46 ^d^	0.470	<0.001	<0.001	<0.001
Day 35	16.80 ^a^	15.44 ^a^	15.32 ^a^	15.15 ^a^	15.60 ^a^	16.35 ^a^	18.60 ^b^	0.272	0.003	<0.001	<0.001

^1^ Values are mean of six replicates per treatment. ^2^ PC: Positive control diet contained the recommended Ca and non-phytate phosphorus; NC-1: negative control diet contained the recommended Ca and 0.10% available P deficiency; NC-2: negative control diet contained the recommended Ca and 0.15% available P deficiency; NC-3; negative control diet contained the recommended Ca and 0.20% available P deficiency; NC-3-500, NC-3-1000, and NC-3-1500 are NC-3 plus phytase 500, 1000, 1500 FTU/kg, respectively. ^3^ Pooled standard error of mean. ^4^ Orthogonal polynomial contrasts were conducted to assess the significance of the linear or quadratic effects of the supplementation with phytase in the broilers. ^a–d^ Values in a row with different superscripts differ significantly (*p* < 0.05).

**Table 6 animals-14-00041-t006:** Effect of phytase inclusion in the diets on digesta IP_6-3_ concentrations in the duodenum/jejunum and ileum of the broiler chickens ^1^.

Period	Dietary Treatment ^2^							SEM ^3^	*p*-Value	Polynomial Contrast ^4^	
	PC	NC-1	NC-2	NC-3	NC-3-500	NC-3-1000	NC-3-1500			Linear	Quadratic
Digesta duodenum/jejunum on day 21 (µM/g)											
IP_6_	0.791 ^c^	0.702 ^c^	0.726 ^c^	0.687 ^c^	0.503 ^b^	0.359 ^a^	0.404 ^ab^	0.030	<0.001	0.001	0.001
IP_5_	0.398	0.375	0.484	0.345	0.275	0.343	0.258	0.027	0.350	0.540	0.828
IP_4_	0.097 ^a^	0.123 ^ab^	0.127 ^ab^	0.130 ^ab^	0.192 ^b^	0.267 ^b^	0.217 ^b^	0.013	0.002	0.037	0.035
IP_3_	0.026 ^ab^	0.005 ^a^	0.015 ^a^	0.017 ^a^	0.047 ^b^	0.077 ^c^	0.145 ^d^	0.008	<0.001	<0.001	<0.001
Digesta ileum on day 21 (µM/g)											
IP_6_	0.661 ^b^	0.667 ^b^	0.668 ^b^	0.651 ^b^	0.322 ^a^	0.255 ^a^	0.229 ^a^	0.036	<0.001	<0.001	<0.001
IP_5_	0.388	0.316	0.362	0.290	0.217	0.209	0.195	0.025	0.229	0.115	0.228
IP_4_	0.111	0.130	0.121	0.141	0.202	0.158	0.192	0.011	0.183	0.442	0.683
IP_3_	0.0258 ^a^	0.0218 ^a^	0.0358 ^a^	0.0320 ^a^	0.119 ^b^	0.088 ^b^	0.114 ^b^	0.009	<0.001	0.058	0.078
Digesta duodenum/jejunum on day 35 (µM/g)											
IP_6_	0.689 ^b^	0.638 ^b^	0.604 ^b^	0.657 ^b^	0.359 ^a^	0.356 ^a^	0.351 ^a^	0.027	<0.001	<0.001	<0.001
IP_5_	0.315	0.315	0.299	0.302	0.240	0.264	0.222	0.016	0.566	0.303	0.583
IP_4_	0.124	0.110	0.118	0.128	0.167	0.164	0.182	0.009	0.187	0.116	0.268
IP_3_	0.0097 ^a^	0.0083 ^a^	0.0153 ^a^	0.0093 ^a^	0.214 ^b^	0.176 ^b^	0.167 ^b^	0.0181	<0.001	0.058	0.013
Digesta ileum on day 35 (µM/g)											
IP_6_	0.676 ^c^	0.684 ^c^	0.620 ^c^	0.606 ^bc^	0.448 ^ab^	0.433 ^a^	0.423 ^a^	0.027	0.005	0.083	0.133
IP_5_	0.332 ^bc^	0.349 ^c^	0.403 ^c^	0.357 ^c^	0.242 ^ab^	0.224 ^a^	0.199 ^a^	0.016	0.001	<0.001	<0.001
IP_4_	0.0913 ^a^	0.103 ^a^	0.0907 ^a^	0.103 ^a^	0.212 ^b^	0.190 ^b^	0.190 ^b^	0.0098	<0.001	0.014	0.001
IP_3_	0.0183 ^a^	0.0277 ^a^	0.0078 ^a^	0.0441 ^a^	0.133 ^b^	0.147 ^b^	0.106 ^b^	0.0111	<0.001	0.152	0.029

^1^ IP_6-3_ (IP_6_ = Myo-inositol hexa-phosphates, IP_5_ = Myo-inositol penta-phosphates, IP_4_ = Myo-inositol tetra-phosphates, and IP_3_ = Myo-inositol tri-phosphates): the means calculated from six replicates per treatment. ^2^ PC: Positive control diet contained the recommended Ca and non-phytate phosphorus; NC-1: negative control diet contained the recommended Ca and 0.10% available P deficiency; NC-2: negative control diet contained the recommended Ca and 0.15% available P deficiency; NC-3: negative control diet contained the recommended Ca and 0.20% available P deficiency; NC-3-500, NC-3-1000, and NC-3-1500 are NC-3 plus phytase 500, 1000, 1500 FTU/kg, respectively. ^3^ Pooled standard error of mean. ^4^ Orthogonal polynomial contrasts were conducted to assess the significance of the linear or quadratic effects of the supplementation with phytase in the broilers. ^a–d^ Values in a row with different superscripts differ significantly (*p* < 0.05).

**Table 7 animals-14-00041-t007:** Regressions of the concentrations of tibia phosphorus on available phosphorus from the diets and phytase supplementation.

Items	Day 21				Day 35			
	Regression Equation							
	Linear	R^2^	Quadratic	R^2^	Linear	R^2^	Quadratic	R^2^
Available phosphorus (AP)								
Tibia P (%)	Y = 5.80 + 25.37X_AP_	0.9542	Y = 15.80 − 33.90X_AP_ + 83.82X_AP_^2^	0.9916	Y = 13.11 + 8.36X_AP_	0.8876	Y = 17.83 − 22.53X_AP_ + 47.73X_AP_^2^	0.9915
Phytase								
Tibia P (%)	Y = 12.87 + 0.005X_phy_	0.9283	Y = 12.99 + 0.004X + (4.6 × 10^−7^)X_phy_^2^	0.9300	Y = 14.76 + 0.002X_phy_	0.8748	Y = 15.21 − 0.0005X_phy_ + (1.800 × 10^−6^)X_phy_^2^	0.9898

Y = predicted performance for a given criteria; X_AP_ = percentage of AP from the diet; X_phy_ = FTU of phytase per kg.

**Table 8 animals-14-00041-t008:** Equivalency of available phosphorus (AP) and equivalency values (%) of phytase.

Items		Equivalency of AP Values (FTU/kg) ^1^		
		500	1000	1500
Calculated matrix value (%) ^2^				
Linear equation	Day 21	0.377	0.476	0.574
	Day 35	0.317	0.437	0.557
Quadratic equation	Day 21	0.383	0.448	0.504
	Day 35	0.307	0.403	0.500

^1^ Equivalency of AP values (%) relative to NC-3 diet. ^2^ Calculated values were calculated by substituting the phosphorus percent in the tibia to estimate the equivalency available phosphorus of phytase.

## Data Availability

Data are contained within the article.

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
