# Peer review of "Efficacy and Equivalency of Phytase for Available Phosphorus in Broilers Fed an Available Phosphorus-Deficient Diet"

_animals, 2023, doi:10.3390/ani14010041_

Round 1

Reviewer 1 Report

Comments and Suggestions for Authors

The manuscript by Yu et al describes the role of phytase on broiler performance and nutrient digestibility when birds are fed diets deficient in available phosphorus (AP). The paper is well written, but the findings are a well-established fact that have emerged from past search studies.  

Minor Issues

Introduction

Ln. 73. Correct the sentence to read “The study also investigated the AP equivalency of phytase….”

Material and Methods

2.2 Birds and housing

Ln.84 replace “weighted” with “weighed”.

Light schedule used during this experiment is not provided.

2.4. Growth performance measurements

Ln. 131. Replace “disappearance” with “consumption”.

2.6. Nutrient digestibility

One aspect of the study as mention in the introduction was to reduce P excretion… with this in mind was P measured in the fecal content from different treatments …. this information if available will be a great addition to the manuscript.  

Ln. 154. Replace “disappearance” with “digestibility”.

4. Discussion.

Ln. 349. Use “The” instead of “These”.

Ln. 354-355. Rewrite this sentence to read” ……addition of phytase to the diet and different levels of available phosphorus content ….”

Ln. 362. Use “the” instead if “this”

Ln 397-401.  This segment of the discussion can be expounded to help your reader understand what’s the role of ileal pH on availability of phosphorus and its interaction with Zn.

5. Conclusion.

Ln. 404-405. This sentence is not clear and needs to be corrected.

Comments on the Quality of English Language

No comments!

Author Response

Thank you for your consideration of my research paper.

Reviewer 2 Report

Comments and Suggestions for Authors

Review for the paper titled "Efficacy and Equivalency of Phytase for Available Phosphorus in Broilers Fed an Available Phosphorus-Deficient Diet"

Abstract

The sentence in lines 29-33 requires rephrasing for enhanced clarity. It is currently convoluted and challenging to comprehend.

Introduction

The introduction segment lacks an adequate presentation of the underlying issue. Incorporating references to pertinent studies, and their findings, and elucidating what distinguishes this study's relevance from others would significantly strengthen this section.

Materials and Methods

Line 91: Specify the temperature and light program more explicitly.

References 12 and 14: Verify if they are the same.

Clarify whether the experiment commenced at the start of the starter phase (1 day) or from the grower phase (14 days). Also, eliminate confusion in notation, particularly between the PC group and the three experimental groups (NC-1, NC-2, and NC-3), alongside the confusing NCP1, NCP2, and NCP3 designations.

Address the discrepancy between the duration of the starter phase mentioned in the text (1-14 days) and that indicated in Table 1 (21 days), alongside the initiation of the grower phase.

Specify the measurement units for each parameter (e.g., BW in grams, ADG in grams/day).

Results

Clarify how "ileal" nutrient digestibility was determined if the digesta from the duodenum/jejunum and ileum were collected in the same tubes.

Properly cite AOAC as a reference (line 168).

Discussion

Elaborate on the poor production performances, particularly explaining the significant increase in the FCR of the NC-3 group. Provide a detailed explanation of the adverse effects of the diet and explore this aspect further, especially considering the contradictory results observed in the NCP-3 group. Merely confirming the agreement of results with other studies isn't sufficient; delve deeper into potential gaps in the field that this study could address.

Explore in-depth how the inadequacy of available phosphorus can impact bone development and articulate key points highlighting the importance of ensuring this nutrient for broilers to prevent nutritional imbalances.

Avoid repeatedly mentioning that a certain aspect has been extensively studied, as this might suggest the study lacks novelty. Instead, focus on presenting more substantial statements in the discussion, moving beyond mere comparisons with existing data and assumptions. Expand the discussion to offer comprehensive insights.

Author Response

(The authors gave the same response as above.)

Round 2

Reviewer 2 Report

Comments and Suggestions for Authors

The authors corrected the paper as indicated in the first round. I have no further observations.